# The Interplay of Likeability and Fear in Willingness to Pay for Bat Conservation

**Vasilios Liordos \***, **Vasileios J. Kontsiotis, Orestis Koutoulas** and **Aristarchos Parapouras**

Department of Forest and Natural Environment Sciences, International Hellenic University, P.O. Box 172, 66100 Drama, Greece; vkontsiotis@for.ihu.gr (V.J.K.); orestis1982@hotmail.com (O.K.); aris.parapouras@gmail.com (A.P.)
\* Correspondence: liordos@for.ihu.gr

**Abstract:** Bats populations and their habitats are currently threatened globally, but particular declines have been seen across Europe. The contingent valuation method is commonly used to assign an economic value to species conservation through a willingness to pay (WTP) surveys. We carried out face-to-face interviews of a representative to the Greek population sample (*n* = 1131) and used a multiple-bounded discrete choice approach to estimate WTP for bat conservation. More than half of the Greek population was supportive of bat conservation (54.6%). Mean WTP was estimated at €21.71, and the total amount that could be collected was €105.6 million, after considering the number of taxpayers and the proportion of supportive people. There was an interplay between emotions, with likeability being the most important positive predictor of support, and fear the most important negative predictor of WTP for bat conservation. Among sociodemographics, older participants, with higher education, farmers, and pet owners showed the highest support, while those with higher education, farmers, and consumptive recreationists offered the highest bid for bat conservation. Participants drew information about bats mostly from informal sources, such as friends, movies, novels, and comics. Our study allowed for the estimation of public support and necessary funds for bat conservation, which are valuable for successful conservation management. Findings will also be critical for the design and implementation of effective education and outreach programs, aimed at increasing knowledge about bats and ultimately support for bat conservation actions.

**Keywords:** Chiroptera; questionnaire survey; Greek public; contingent valuation; conservation; emotions; sociodemographics



## 1. Introduction

Human behavior and activities have inflicted profound negative impacts on Earth, being responsible for the exceptionally rapid loss of animal species over the last few centuries [1]. The estimated average rate of vertebrate species loss over the last century is up to 100 times higher than the pre-human background rate, indicating that a sixth mass extinction is already underway [1]. Currently, thousands of animal species populations are threatened with extinction [2].

Among mammals, bats are the second largest order (Chiroptera), with more than 1400 species, widely dispersed across six continents [3]. Bats are essential to the health of global ecosystems, providing vital ecosystem services, such as insect pest consumption, plant pollination, and seed dispersal [4]. However, they face unprecedented threats mostly from the ongoing habitat degradation and destruction, the accelerated climate change, and the expansion of wind parks [5]. As a result, about 16% of the world's bat species are threatened (classified as critically endangered, endangered, or vulnerable), 7% are near threatened, while about 18% are data deficient, indicating that more conservation attention is necessary for these species [6]. The urgent need for the protection of threatened bat species has been recognized, and many governmental and non-governmental conservation programs have been implemented or are in progress, worldwide [7]. Among the factors

necessary for the success of bat conservation programs, knowing the public's willingness to support them and timely securing required funds are among the most important [7,8]. Being implemented either by governmental or by non-governmental organizations, conservation programs rely heavily on governments or supra-governmental institutions for funding. Taxpayers represent the biggest source of revenue for governments. Therefore, knowing the public's willingness to support conservation programs and estimating the amount they would be willing to pay is critical for the successful conservation of bat species.

It is difficult to assign direct economic values to ecosystem services provided by wildlife species, and even more difficult to evaluate the cost of averting endangerment and restoring healthy populations [9]. Therefore, methods assigning indirect values to benefits provided by wildlife species have been used [9,10]. Stated preference methods, mostly the contingent valuation method (CVM), are commonly used to assign an economic value to species or groups of species through a willingness to pay (WTP) for their conservation surveys (e.g., Gray Wolf *Canis lupus* [11], Tiger *Panthera tigris* [12], Finless Porpoise *Neophocaena phocaenoides* [13], Loggerhead Sea Turtle *Caretta caretta* [14], whales [15], White Stork *Ciconia ciconia* [16], Golden-cheeked Warbler *Setophaga chrysoparia* [17], birds [18], Corncrake *Crex crex* [19]. To date, there are few studies concerning the WTP for the conservation of bat species in the literature. Two studies concerned the WTP for the conservation of fruit-eating bat species (Ryukyu Flying Fox *Pteropus dasymallus* in Japan [20], Mauritian Flying Fox *Pteropus niger* in Mauritius [21]), and one study concerned insect-eating bat species (Mexican Free-tailed Bat *Tadarida brasiliensis mexicana* in Mexico and the U.S. [22]). WTP surveys are helpful in determining public support for species conservation and in identifying potential funding sources for conservation programs [9].

Emotions, participation in outdoor recreational activities, and sociodemographic characteristics are among the most important predictors of support and WTP for wildlife species conservation [8,23–25]. The importance of living organisms for people has been elaborated by the writings of E.O. Wilson who maintained that we gain the most satisfaction from processes that mimic the nature of life on many levels, defining biophilia as our "innate tendency to focus on life and lifelike processes" [26], also including a biophobic component; learned negative "avoid" behaviors [27]. Zajonc [28] argued for the primacy of affect over cognition, citing that "we can like something or be afraid of it before we know precisely what it is and perhaps even without knowing what it is". Jacobs et al. [24] proposed that the study of emotions is vital for enhancing the understanding of human–wildlife interactions. Likeability and fear are among the most studied emotions of humans towards wildlife. Higher support and WTP for their conservation have been reported for the animal species perceived as more likeable and less fearsome than those perceived as less likable and more fearsome (e.g., [8,23,29]). Bats are classified among the least likeable and more fearsome animal groups, along with frogs, spiders, rats, and snakes [8,23,30], with likeability being positively and fear negatively associated with support and WTP for their conservation [20,21]. In a study of Greek people, Liordos et al. [8] reported that a bat species, the western barbastelle *Barbastella barbastellus*, was the least likeable among 12 endangered species, including four mammals, three birds, two reptiles, one fish, one amphibian, and one arachnid. They found that 70% of the survey participants disliked the western barbastelle, while only 15% liked it. Also, the survey participants considered the bat rather fearsome, with 44% of them feeling safe and 36% feeling afraid around the species. In a study of Slovakian students, Prokop et al. [30] reported that only 9% of the students wanted to have bats in the loft of their homes, 65% did not want to camp near places inhabited by bats, 57% did not want to catch a bat in their hands, while 34% said that even the thought of touching a bat scared them.

Wildlife-related outdoor recreational activities refer to the direct experience and exploration of wildlife and can be categorized as consumptive and non-consumptive. Consumptive recreation involves the harvest of wildlife, as in hunting and fishing. Non-consumptive recreation includes non-extractive activities, such as wildlife-watching and

photography. Cooper et al. [31] found that consumptive recreationists, such as hunters, and non-consumptive recreationists, such as birdwatchers, were 4–5 times more likely than non-recreationists to participate in conservation activities. Hunters are advocates of wildlife management, especially helping to conserve game animals and their habitat [32,33], but they are usually negative towards the protection of species they perceive as a threat to their game or hunting dogs [34–36]. Outdoor recreation categories have not been used as predictors of WTP for wildlife conservation. However, bats do not negatively impact fishing or hunting practices [4].

Among sociodemographic characteristics, age, gender, income, level of education, place of residence, occupation, pet ownership, and participation in outdoor recreation have been proposed as important factors predicting support for wildlife species conservation. Richer, more educated, younger females were more willing to support and pay for the conservation of the Mexican Free-tailed Bat in Mexico and the U.S. [22]. Age, gender, income, level of education, and place of residence were not significant predictors of WTP for the conservation of the Ryukyu Flying Fox in Japan [20] and the Mauritian Flying Fox in Mauritius [21]. Occupation and pet ownership have not been used as predictors of WTP for wildlife conservation. Farmers usually support conservation and management actions that do not negatively affect their crops [37]. The insect-eating bats of Greece do not harm crops; in fact, they are highly beneficial by consuming large numbers of insect pests [4]. Pet owners tend to be more effective towards wildlife species and more supportive of their conservation than non-pet owners [38,39].

Greece supports a high diversity of bats, hosting 35 of the 43 European species, all insect-eating. About 23% of the bat species have been assigned to threat status and another 23% have been classified as near threatened in the Red Data Book of Greece [40]. Also, all the Greek bat species have been classified as species whose conservation requires the designation of special areas of conservation (about 31%) and/or in need of strict protection (all species) by the European Union's (E.U.) Habitats Directive, and as either strictly protected fauna species (97%) or protected fauna species (3%) by the Bern Convention. However, studies concerning the WTP for the conservation of bat species in Greece or Europe are not available in the literature. Furthermore, conservation and management plans are currently designed in Greece, primarily concerning the protection of very important wildlife species, including bats (Annex II species in Habitats Directive) and their habitats, in designated special areas of conservation (SAC; Natura 2000 sites). The disturbance of colonies by cave visitors, the destruction of mature trees and abandoned buildings, aridity, and forest fires that reduce the availability of insect prey, are the most important threats that bats are currently facing in Greece [40]. Funds for the design and implementation of conservation and management plans are jointly provided by the E.U. and the Greek government. Greek funds are mostly taxpayers' money. Knowing the proportion of the Greek population that is willing to contribute to the conservation of bat species and the corresponding monetary WTP would allow for determining their degree of support for bat conservation and for identifying potential sources for its funding.

Our first objective was, therefore, to estimate the WTP of the Greek population with the CVM using a multi-bounded discrete choice approach [41]. Next, we aimed at estimating the power of emotions (likeability, fear) and sociodemographics (age, gender, income, level of education, place of residence, occupation, pet ownership, participation in outdoor recreation) as predictors of WTP. Based on previous findings from the literature and our objectives, we hypothesized that participants will have a moderate WTP for bat conservation.

## 2. Materials and Methods

### 2.1. Sampling Protocol

The study was carried out in the 13 administrative Regions of Greece, with a population of 10,816,286 people [42] (Figure 1), 8,931,570 of which submitted a tax return form in 2020 [43]. Data was collected from on-site face-to-face interviews with adult Greek resi-

dents (aged 18–80), between March 2017 and September 2018. The capital city, two towns, and three villages were visited in each of the 52 Prefectures in all the Regions, in an effort to collect a geographically representative sample. Interviews were carried out during open market hours (9.00–15.00 and 17.00–21.00, from Monday to Saturday). Every fifth person passing in front of the researcher (O.K.) was asked to participate by completing a questionnaire. In cases in which more than five persons had passed while a questionnaire was being completed, the first person encountered upon completion was selected [44]. It took respondents 15 min on average to complete the questionnaire.

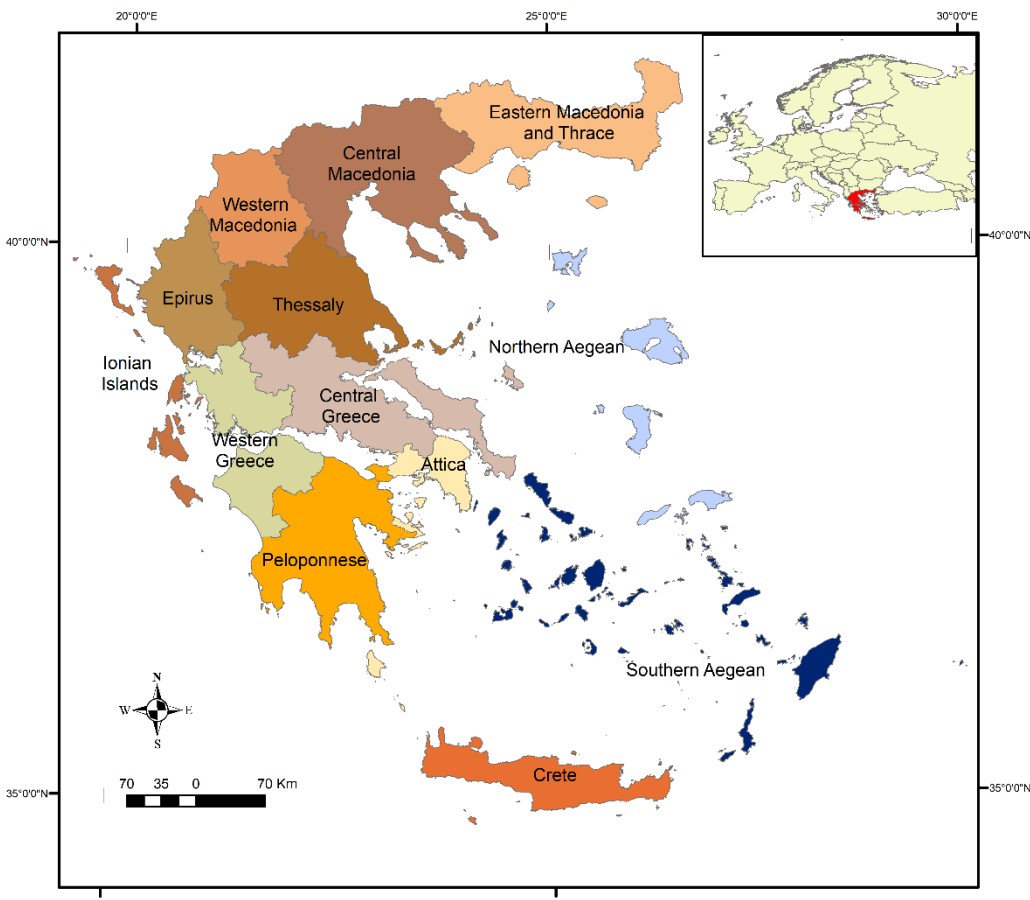

**Figure 1.** Map showing the 13 administrative regions of Greece, where the survey was carried out.

*2.2. Survey Design*

The questionnaire consisted of two parts. The first part included the WTP questions. The estimation of WTP involved a two-step procedure. In the first step, participants were asked "Many of the populations of the 35 bat species of Greece are vulnerable and threatened with extinction, mainly due to the degradation or destruction of mature forests and caves. Would you support a governmental program to protect bats and their habitats by paying an amount annually for the next five years?" Possible answers were "yes" and "no." The second step included only the participants that replied yes in the first step: "Below are listed several amounts of an annual tax that you will have to pay for the next five years for the conservation of bat species and their habitats. Check for each amount how certain you are about paying that amount." The amounts were €1, €5, €10, €20, €40, €80, €150, €300, and €500, and possible answers were "definitely yes," "probably yes," "not sure," "probably no" or "definitely no". We offered a wide range of choices, from a minimal to a considerable amount, as proposed by Johansson et al. [29] and Broberg and Brännlund [45]. The annual tax is applicable on the individual level, because tax return forms are submitted separately for each person, and not on a household basis in Greece [43].

The second part of the questionnaire included questions about variables used to explain WTP in subsequent modeling: (a) sociodemographic factors, such as age, gender, level of education, income, current residence, occupation, pet ownership, participation in consumptive and non-consumptive recreation activities, and (b) emotions, such as likeability and fear of bats. The definition of these variables and their measurement scales are given in Table 1. Also, in this part, the participants were asked to identify and report their sources of knowledge about bats. They were presented with the following options: the movies, fiction (novels, comics), non-fiction (textbooks, documentaries), the news, friends, school, none, others, and were allowed to select more than one option.

**Table 1.** Variables used in the willingness to pay (WTP) for bat conservation analysis.

| Variable | Definition | Mean | SD | Min | Max |
|---|---|---|---|---|---|
| WTP | 1 if the participant is willing to pay for the conservation of bat species in Greece. | 0.55 | 0.50 | 0 | 1 |
| Bat likeability | How strongly the participant likes or dislikes bats (1 = strongly dislike, 2 = dislike, 3 = neither, 4 = like, 5 = strongly like). | 2.32 | 0.92 | 1 | 5 |
| Bat fear | How safe or afraid would the participant feel if encountering bats (1 = very safe, 2 = safe, 3 = neither, 4 = afraid, 5 = very afraid). | 2.64 | 1.21 | 1 | 5 |
| Age | Years of age. | 46.69 | 15.37 | 18 | 85 |
| Gender | 1 if the participant is a woman. | 0.53 | 0.50 | 0 | 1 |
| Level of education | 1 if lower (elementary or high school degree), 2 if higher (technological institute or university degree). | 1.25 | 0.45 | 1 | 2 |
| Income | Participant's household income (€ × 1000). | 11.85 | 18.14 | 0 | 250 |
| Residence | 1 if the participant lives in a rural area, 0 if the participant lives in an urban area. | 0.23 | 0.41 | 0 | 1 |
| Occupation | 1 if the participant is a farmer, 0 if the participant is not a farmer. | 0.11 | 0.32 | 0 | 1 |
| Pet ownership | 1 if the participant owns a pet, 0 if the participant does not own a pet. | 0.59 | 0.49 | 0 | 1 |
| Consumptive recreation | How often the participant goes for hunting or fishing (1 = never, 2 = rarely, 3 = sometimes, 4 = often, 5 = very often). | 2.05 | 0.88 | 1 | 5 |
| Non-consumptive recreation | How often the participant is involved in outdoor activities other than hunting and fishing, e.g., wildlife-watching and photography (1 = never, 2 = rarely, 3 = sometimes, 4 = often, 5 = very often). | 2.57 | 1.00 | 1 | 5 |

*2.3. Econometric Model*

The econometric models used derive from a random utility framework. The first is a simple choice model (yes/no) on the probability of paying for the implementation of a bat conservation program. It is a binary logistic model [10] with the WTP (yes = 1, no = 0) as

the dependent variable and likeability, fear, and the sociodemographic variables of Table 1 as the independent variables.

The second model included only the participants who answered yes in the first, yes/no, model. The data for the second model were in a multiple-bounded payment card format (i.e., participants recorded their degree of certainty for paying each amount presented to them). We used the interval model proposed by Welsh and Poe [41] for analyzing multi-bounded discrete choice data. We adopted the "probably yes" approach for recoding the data, such that "definitely yes" and "probably yes" mean "yes,", and "not sure," "probably no," and "definitely no" mean "no." This approach gives results similar to other standard discrete choice models, such as dichotomous choice, payment card, and open-ended [41]. After recoding the multiple-bounded data into terms of "yes" and "no", it can be treated in the same way as ordinary double-bounded discrete choice data [45,46]. Under the interval model approach, the highest bid the participants accept and the lowest bid that they do not accept are considered. If $A^L$ is the highest "yes" bid, and $A^U$ the lowest "no" bid, the maximum WTP is $A^L \leq WTP < A^U$. Then, given a distribution function $F$ for WTP, the likelihood is [41,45,46]:

$$\ln L = \sum_{i=1}^{N} [\ln(F(A^U) - F(A^L)]$$  (1)

And assuming a log-logistic distribution:

$$F(A^U) = \left(1 + e^{\delta X - \alpha \ln(A_i^U)}\right)^{-1}$$  (2)

and

$$F(A^L) = \left(1 + e^{\delta X - \alpha \ln(A_i^L)}\right)^{-1}$$  (3)

where $X$ is the vector of covariates, and $\delta$ is the corresponding parameter vector. The parameter $\alpha$ corresponds to the bid and can be interpreted as the marginal utility of money. Mean WTP is then calculated as:

$$MWTP = e^{\frac{\delta X}{\alpha} + \left(\frac{\alpha^{-1}}{2}\right)^2}$$  (4)

### 2.4. Data Analysis

We dealt with multicollinearity issues by including in the modeling process only variables with low variance inflation factors (VIF < 5;) and correlations ($r_s < 0.7$). VIFs were calculated using the function *vifstep* of the *usdm* R package [47] and correlations using the function *cor.test* of the *ggpubr* R package [48]. All VIFs were < 1.7 and correlations < 0.400, therefore all variables were included in the initial models.

The binary logistic regression, yes/no, model was fitted using the function *glm* (a binomial distribution with logit link function) of the *stats* R package [49]. The full model was fitted first, and then the final model, containing only the significant variables, was estimated with forward selection. Odds ratios were calculated using the *logitor* and marginal effects at the mean using the *logitmfx* functions of the *mfx* R package [50].

The second, highest yes/lowest no bids, model was fitted with a log-logistic distribution using the function *dbchoice* of the *DCchoice* R package [51]. Confidence intervals (95% CI) for the mean WTP were calculated with the nonparametric bootstrap method using the *bootCI* function. The initial full model and the final model containing only the significant variables were estimated.

All statistical analyses were performed in program R 4.0.2 [49]. The significance level was set at $\alpha = 0.05$.

## 3. Results

### *3.1. Sociodemographic Variables, Likeability and Fear*

A total of 1131 questionnaires were completed, with 111 refusals, yielding a response rate of 90.1%. Participants replied to all the questions. Greece's population has a 50.8% female/49.2% male gender ratio (53.1%/46.9% in this study), the age ratio, after excluding those under 18 and over 80, is 28.5%/37.1%/34.4% (30.2%/38.1%/31.7% in this study) in the 18–34, 35–54, and 55+years old age classes, respectively, the lower/higher educational level ratio is 73.4%/26.6% (75.1%/24.9% in this study) [52] and the rural/urban ratio is 21.0%/79.0% (23.1%/76.9% in this study) [53]. The sample's gender, age, educational level and current residence (rural/urban) ratios were not different to that of the population's (gender: $\chi^2 = 1.245$, df = 1, $p = 0.252$; age: $\chi^2 = 3.683$, df = 2, $p = 0.159$; educational level: $\chi^2 = 1.517$, df = 1, $p = 0.206$; current residence: $\chi^2 = 2.816$, df = 1, $p = 0.086$).

Farmers consisted of 11% of the sample, while 59% owned a pet (Table 1). A high proportion of the participants (85%) reported that they were never or rarely involved in consumptive activities, such as hunting and fishing, while only 10% of the participants were involved in consumptive activities often or very often (mean 2.05 ± 0.88). On the other hand, 51% of the participants were never or rarely involved in non-consumptive outdoor activities, such as bird-watching and wildlife photography, while 22% of the participants were involved in non-consumptive activities often or very often (2.57 ± 1.00).

In general, survey participants reported low levels of likeability (2.32 ± 0.92) and more moderate levels of fear (2.64 ± 1.21) (Table 1). Only 10% of the participants liked or strongly liked bats, 59% of the participants disliked or strongly disliked bats, while 31% of the participants neither liked nor disliked bats. Almost half of the participants (48%) would feel safe if encountered bats, 25% of the participants would feel afraid or very afraid around bats, while 27% of the participants were neither afraid nor feeling safe around bats.

The survey participants learned about bats primarily from the movies (31.7%; Figure 2). Other important sources of knowledge were their social circle (21.0%), non-fiction (textbooks and documentaries; 16.2%), and the news (15.4%). An important proportion of the participants did not know anything about bats, as 19.1% of them did not identify any source of information.

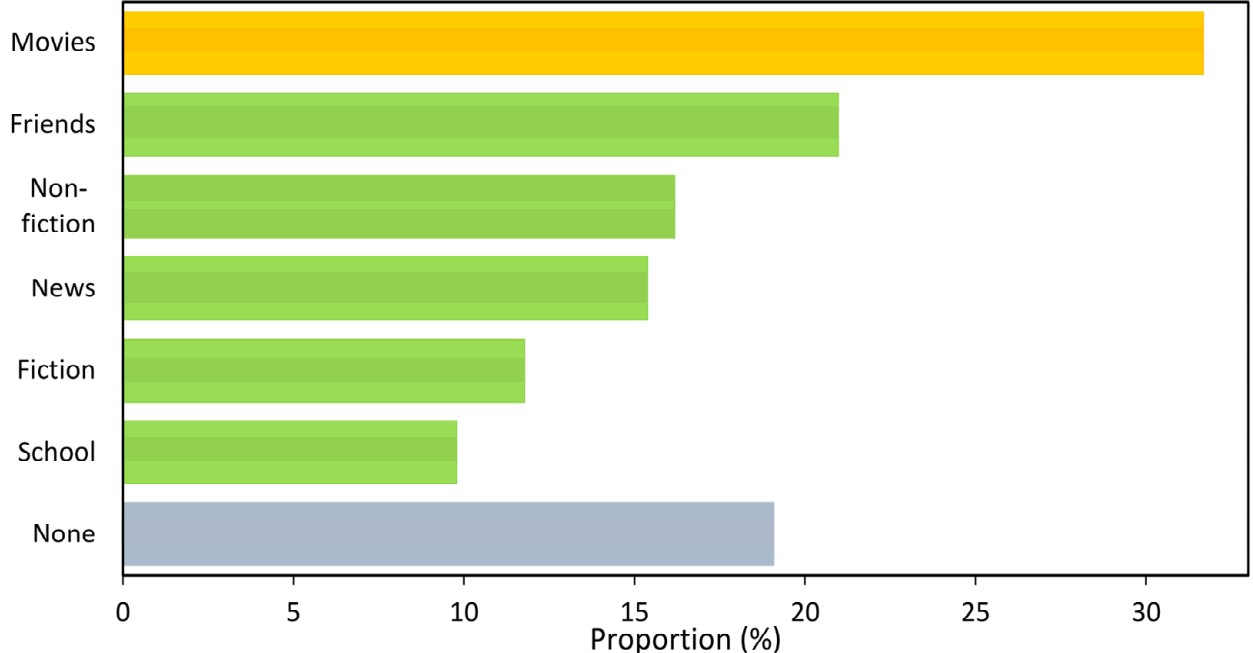

**Figure 2.** Sources from which the survey participants reported that they draw factual knowledge about bats. Category "Non-fiction" refers to textbooks, documentaries and special press. Category "Fiction" refers to novels and comics.

### 3.2. WTP Models

Among the participants, 618 (54.6%) were willing to pay for the conservation of bat species (Table 1). The proportion of correct predictions for the whole sample was 73.1% for the initial and 71.4% for the final binary logistic regression, yes/no, model (Table 2). Among emotions, those who liked bats more were more willing to pay for the conservation of bat species than those who liked bats less ($p < 0.001$). When all independent variables are at their mean values, the expected probability of WTP will increase by 0.234 with a marginal increase in bat likeability.

**Table 2.** Results of the binary logistic regression willingness to pay model (yes/no, $n = 1131$).

| Variable | Full Model | | | Final Model | | |
|---|---|---|---|---|---|---|
| | Odds Ratio | Marginal Effects | *p* | Odds Ratio | Marginal Effects | *p* |
| Bat likeability | 2.314 | 0.205 | <0.001 | 2.534 | 0.234 | <0.001 |
| Bat fear | 0.962 | −0.009 | 0.603 | | | |
| Age | 1.025 | 0.006 | <0.001 | 1.025 | 0.006 | <0.001 |
| Gender (Female) | 1.064 | 0.015 | 0.734 | | | |
| Level of education (Higher) | 1.911 | 0.156 | <0.001 | 1.993 | 0.166 | <0.001 |
| Income | 1.000 | 0.000 | 0.676 | | | |
| Residence (Rural) | 0.830 | −0.046 | 0.283 | | | |
| Occupation (Farmer) | 1.899 | 0.151 | 0.016 | 1.913 | 0.157 | 0.011 |
| Pet ownership (Yes) | 1.416 | 0.086 | 0.035 | 1.454 | 0.095 | 0.025 |
| Consumptive recreation | 0.956 | −0.011 | 0.654 | | | |
| Non-consumptive recreation | 1.083 | 0.020 | 0.312 | | | |
| Nagelkerke's $R^2$ | | 0.217 | | | 0.211 | |
| −LogLik | | 488.774 | | | 490.103 | |
| AIC | | 1001.500 | | | 992.210 | |

Among the sociodemographic factors, older participants ($p < 0.001$), with higher education ($p < 0.001$), farmers ($p = 0.011$), and pet owners ($p = 0.025$) were more willing to pay for the conservation of bat species than younger participants, those with lower education, non-farmers, and non-pet owners (Table 2). When all independent variables are at their mean values, the expected probability of WTP will increase by 0.006, 0.166, 0.157, and 0.095 with a marginal increase in age, the proportion of the highly educated, the proportion of farmers, and the proportion of pet owners, respectively.

In the second-highest yes/lowest no bids model, we estimated how much those participants who had replied yes in the first model were willing to pay for bat conservation (Table 3). Among emotions, those who feared bats more were willing to pay a lower amount for the conservation of bat species than those who feared bats less ($p < 0.001$). Among the sociodemographic factors, participants with higher education ($p = 0.005$), farmers ($p = 0.023$), and those who participated more in consumptive recreation activities ($p = 0.005$) were willing to pay a higher amount for the conservation of bat species than participants with lower education, non-farmers and those who participated less in consumptive recreation activities.

The mean amount of WTP was estimated at about €21.71 (95% CI: 19.9–23.9). Considering the mean value, confidence intervals, the proportion of the participants who were willing to pay for bat conservation, and the number of taxpayers in Greece, the amount of €105.6 million (min €96.4 million, max €115.7 million) could be collected in taxes for the conservation of bat species.

**Table 3.** Results of the log-logistic regression willingness to pay model (highest yes/lowest no bids, *n* = 618).

| Variable | Full Model | | Final Model | |
|---|---|---|---|---|
| | Coefficient | *p* | Coefficient | *p* |
| Bat likeability | 0.078 | 0.474 | | |
| Bat fear | −0.674 | <0.001 | −0.779 | <0.001 |
| Age | −0.003 | 0.667 | | |
| Gender (Female) | 0.208 | 0.273 | | |
| Level of education (Higher) | 0.637 | 0.007 | 0.715 | 0.005 |
| Income | 0.076 | 0.439 | | |
| Residence (Rural) | 0.243 | 0.241 | | |
| Occupation (Farmer) | 0.622 | 0.033 | 0.681 | 0.023 |
| Pet ownership (Yes) | 0.115 | 0.568 | | |
| Consumptive recreation | 0.371 | 0.007 | 0.452 | 0.005 |
| Non-consumptive recreation | 0.108 | 0.237 | | |
| Nagelkerke's $R^2$ | 0.244 | | 0.229 | |
| −LogLik | 560.405 | | 563.610 | |
| AIC | 1146.810 | | 1139.221 | |
| Mean WTP (€) | 21.64 | | 21.71 | |
| 95% CI of mean WTP (€) | 19.747–23.699 | | 19.945–23.900 | |

## 4. Discussion

### 4.1. WTP for Bat Conservation

Our results indicated that more than half of the participants were supportive of bat conservation, as they would fund the implementation of a bat conservation program for a period of five years. The mean annual WTP was moderate, however, a large total amount of money could be collected annually. Our sample's mean annual income was considerably lower than Greece's mean income. We argue that our results could be used to make inferences for the country's population because WTP did not vary with income in our sample. Although the funds required for the conservation of bats in Greece are not known, those that could be collected are believed to be many times higher. This suggests that the benefit/cost ratio of bat conservation is likely large and that a new tax, even if it represents a fraction of the underlying economic value that the Greek residents expressed, could be sufficient to address current conservation needs. These funds could be allocated to various conservation actions, such as the protection of bat populations and their habitats, public education, and outreach. In doing so, a brighter future could be secured for this group of animals. Jaunky et al. [21] reported that 49% of the survey participants accepted to pay for the conservation of the Mauritian Flying Fox in Mauritius, a proportion similar to the Greek samples. U.S. residents were willing to pay €25.4 and Mexican residents €6.8 annually for the conservation of the Mexican Free-tailed Bat in their countries [22]. The residents of Japan were willing to pay annually 11.5 for the conservation of the Ryukyu Flying Fox [20], while the residents of Mauritius would pay €6.0 annually for the conservation of the Mauritian Flying Fox in Mauritius [21]. While these studies did not report mean income, we can assume that the Greeks were also more willing to pay for bat conservation than the Japanese and Mauritians, considering the GDP per capita in Japan in 2014 and in Mauritius in 2016 (Table 4). Also, the WTP to GDP ratio for the Greek sample was favorably compared to that of most recent studies of WTP for the conservation of various taxa, including mammals, birds, reptiles, and amphibians (Table 4).

**Table 4.** WTP for the conservation of various wildlife taxa. GDP per capita corresponds to the study year.

| Taxon | Mean WTP (€) | GDP per Capita (€) [a] | WTP/GDP | Year of Study | Country | Source |
|---|---|---|---|---|---|---|
| White Stork *Ciconia ciconia* | 117.0 | 13110.7 | 0.00892 | 2018 | Poland | [16] |
| Forest Elephant *Loxodonta africana cyclotis* | 20.9 | 3200.8 | 0.00653 | 2014 | Congo | [54] |
| Andean Condor *Vultur gryphus* | 29.3 | 5274.1 | 0.00556 | 2019 | Ecuador | [55] |
| Giant Panda *Ailuropoda melanoleuca* | 26.3 | 6508.2 | 0.00404 | 2014 | China | [56] |
| African Elephant *Loxodonta africana* | 20.8 | 6906.1 | 0.00301 | 2016 | China | [57] |
| Gray Wolf *Canis lupus* | 71.2 | 50947.9 | 0.00140 | 2017 | U.S.A. | [11] |
| Bats | 21.7 | 16752.4 | 0.00130 | 2018 | Greece | Here |
| White Stork *Ciconia ciconia* | 44.6 | 35348.2 | 0.00126 | 2018 | Israel | [16] |
| Northern Pintail *Anas acuta* | 6.8 | 7411.6 | 0.00092 | 2016 | Mexico | [58] |
| Mexican Free-tailed Bat *Tadarida brasiliensis mexicana* | 6.8 | 7872.2 | 0.00086 | 2017 | Mexico | [22] |
| Large carnivores | 28.4 | 36294.9 | 0.00078 | 2004 | Sweden | [29] |
| Mauritian Flying Fox *Pteropus niger* | 6.0 | 8205.9 | 0.00073 | 2016 | Mauritius | [20] |
| Red-crowned Crane *Grus japonensis* | 4.6 | 6508.3 | 0.00071 | 2014 | China | [59] |
| Unique and Charismatic (Gorilla) | 18.5 | 32832.7 | 0.00056 | 2009 | U.K. | [60] |
| Birds | 28.8 | 52988.9 | 0.00054 | 2011 | Australia | [18] |
| Whales | 34.0 | 63016.3 | 0.00054 | 2018 | Iceland | [15] |
| Mexican Free-tailed Bat *Tadarida brasiliensis mexicana* | 25.4 | 50947.9 | 0.00050 | 2017 | U.S.A. | [22] |
| Northern Pintail *Anas acuta* | 23.7 | 49177.9 | 0.00048 | 2016 | U.S.A. | [58] |
| Non-unique, Charismatic (Lion) | 14.9 | 32832.7 | 0.00045 | 2009 | U.K. | [60] |
| Golden-cheeked Warbler *Setophaga chrysoparia* | 18.2 | 45012.1 | 0.00040 | 2013 | U.S.A. | [17] |
| Ryukyu Flying Fox *Pteropus dasymallus* | 11.5 | 32300.8 | 0.00036 | 2014 | Japan | [21] |
| Unique, Non-Charismatic (Frog, Toad, Lizard, Bird) | 11.4 | 32832.7 | 0.00035 | 2009 | U.K. | [58] |
| Northern Pintail *Anas acuta* | 10.2 | 35865.9 | 0.00028 | 2016 | Canada | [56] |
| Corncrake *Crex crex* | 10.8 | 46009.4 | 0.00023 | 2006 | Ireland | [19] |
| Finless Porpoise *Neophocaena phocaenoides* | 2.0 | 28329.1 | 0.00007 | 2018 | S. Korea | [13] |
| Loggerhead Sea Turtle *Caretta caretta* | 1.7 | 24825.1 | 0.00007 | 2016 | S. Korea | [14] |
| Non-unique, Non-Charismatic (Frog) | 0 | 32832.7 | 0.00000 | 2009 | U.K. | [60] |

[a] Source: Country search in World Bank web site [61].

### 4.2. WTP and Emotions

Our results indicated that survey participants perceived bats as unlikeable and somewhat fearsome. Also, the WTP models revealed that likeability was the strongest positive predictor of the willingness to support bat conservation, and fear was the strongest negative predictor of the amount that participants were willing to pay for bat conservation. These findings support those of previous studies of the public's attitudes towards bats, commonly reporting that people perceive them as unlikeable, ugly, and fearsome (e.g., [8,23,30,62–65]), classifying them among the least likeable and attractive animal species [8,23], although not always (e.g., [66,67]). Furthermore, although bats are generally considered more unlikeable than fearsome, a considerable proportion of the public reports fear of bats, which are often classified as somewhat fearsome [8,23].

Animal phobias are among the most common and persistent of all phobias, with bugs, mice, snakes, and bats comprising 46% of these fears [62]. Seligman [68] proposed the "biological preparedness hypothesis" which suggests that modern humans remain "biologically prepared" through natural selection to learn fears of natural objects and situations that threatened the survival of the human species during the course of evolution. Fear of bats is considered disgust-relevant, along with other species such as rats, mice, cockroaches, snails, eels, worms, frogs, because they are directly or indirectly associated with the spread of disease and infection, or perceived as slimy and dirty, as Davey [69,70] suggested. Although evolutionary mechanisms are responsible for bat phobia, culture further reinforces it. In earlier times, they were mostly perceived as symbols of evil,

witchcraft, and bad luck due to their association with darkness and underground caverns. Also, many legends have their roots in the actual physical features of bats (i.e., the size and shape of their wings, ears, teeth, and noses), perceived as mysterious and ominous. In Greek mythology, bats are associated with the underworld (Persephone and Hades), while the bat is a god of death in Mayan culture [71]. In some cultures, they represented happiness and longevity though (e.g., Persian, Chinese).

In modern times, the fear of bats has been reinforced by the myth portraying bats as dark, evil, bloodsucking monsters. Bat species are either insect or fruit-eating, except for three species, resident to South America, which feeds on blood; the vampire bats [72]. Vampire bats were not responsible for the creation of vampire myths, as these myths existed long before Europeans or the rest of the Old World ever knew of the existence of bats that feed on blood [71]. Instead, they were named after these myths upon their discovery by science. The word "vampire" came from the Slavic *vampir*, meaning "blood-drunkenness" [73]. Vampires were mythic creatures, undead, blood-sucking, that could leave their body at will and travel about like an animal or even as flame or smoke. But it wasn't until 1897 when Bram Stoker [74] wrote his classic novel, *Dracula*, that bats were associated with vampires for the first time.

Aesthetics positively and fear negatively affected support for the conservation of the Mauritian Flying Fox [21]. However, only fear affected, negatively, the WTP [21]. In our survey, an interesting interplay was observed between likeability and fear in WTP for bat conservation. It seemed that likeability, and not fear, was the critical factor for expressing an intent to participate in bat conservation. People would support an animal species that they like and not one that they dislike [8,23]. In the second bid model, fear replaced likeability as a powerful predictor of WTP; it was higher among those who did not fear bats. Only participants intending to support bat conservation were included in the second model. As predicted by the first model, those who liked bats the most were members of the "yes" group. This might well explain likeability's low predicting power of WTP in the second model. In this model, fear was the critical factor, as those feeling unsafe around bats were less willing to pay for their conservation.

### 4.3. WTP and Sociodemographics

Older participants, those with higher education, farmers, and pet owners expressed the highest intent for bat conservation. Those with higher education, farmers, and consumptive recreationists were also willing to contribute the highest amount of money. Highly educated young people showed the highest intent for the conservation of the Mexican Free-tailed Bat [22], while age and education did not affect the WTP for the conservation of the Ryukyu Flying Fox [20] and the Mauritian Flying Fox [21]. Musila et al. [65] reported that older and more educated Kenyans had more positive attitudes towards bats. In general, younger, highly educated people have a greater interest in wildlife and highly support their conservation [8,23]. Our results, in line with those of Musila et al. [66], might be explained by a possibly higher knowledge about and experience with bats of older people, factors that generally increase their acceptability and support for conservation [66,68]. The higher support of farmers for bat conservation, among the participants, might be explained by their appreciation of the animals' help in decreasing insect pest populations, a hypothesis confirmed by other bat studies [75,76]. Pet owners have an interest in all animals and are usually proponents of wildlife conservation [38,39].

Contrary to what was expected, income did not predict WTP for bat conservation. WTP increased with income for the conservation of the Mexican Free-tailed Bat [22], but not for the conservation of the Ryukyu Flying Fox [20] and the Mauritian Flying Fox [21]. According to economic principles, bat conservation did not represent either a "normal good" (where WTP would increase with income) or an "inferior good" (where WTP would decrease with income) for the Greek residents, but it is rather inelastic (i.e., the elasticity of demand is zero), being a good that tends to have the same demand regardless of income [77].

Consumptive recreationists selected the highest bids for bat conservation. They usually support wildlife conservation, except when species in question threatened their pastime (e.g., impact fish, game, equipment, dogs [35–37]. Kellert [78] found that hunters had great knowledge about predatory animal species, while anti-hunters had relatively low knowledge about them (i.e., taxonomy, biology, superstitions, and folk knowledge concerning wildlife). Hunters and fishers participate in outdoor activities frequently, both consumptive and non-consumptive, thus, having the opportunity for a hands-on experience of nature and wildlife and gaining direct knowledge about several aspects of the life history of wildlife species, both game and non-game [79]. Also, experienced hunters are strongly attached to their favorite pastime [80,81]. This increased experience with and knowledge about species of consumptive recreationists was most likely responsible for their increased support for the conservation of bats, a group of species that does not negatively impact their activities [4].

### 4.4. Management Implications

Greek residents expressed moderate support and monetary WTP for bat conservation, which, however, could yield a large amount of money that would allow conservationists to design and implement effective conservation plans. Although funds are deemed more than sufficient, an increase in the public's support for bat conservation would be desirable. Besides the securing of necessary funds, public acceptance is also critical for successful wildlife conservation [82,83].

Lunney and Moon [84] argued that there is public blindness to the reality of bats, their natural history, their ecology, and conservation, and concluded that there is an urgent need for accurate information about bats to be engagingly available to the public. The inclusion in environmental education programs of information about bat benefits for agriculture [85] and presenting an aesthetic stimulus like the Panda Bat *Niumbaha superba* [86] significantly increased positive perceptions towards bats. Also, Straka et al. [87] found that presenting photographs of vulnerable and distressed bats might be an important instrument to temporarily increase people's emotional reactions to bats, their wildlife value orientation, and, ultimately, their support for bat conservation.

The Congress Avenue Bridge in Austin, Texas, represents one of the world's most successful bat conservation stories [88]. Bats began to move into the bridge's crevices when it was remodeled in 1980. Following this colonization, citizens concerned about dangers posed by bats started campaigning for having the bats removed. However, a media campaign from conservationists successfully changed citizens' opinions. As a result, the bridge is currently home to 1.5 million Mexican free-tailed bats *Tadarida brasiliensis* that attract nearly 140,000 visitors each year. Bat-watching has created a considerable source of income for the city and also further promotes bat awareness and conservation through activities such as the creation of a viewing area with educational kiosks, bat-watching excursions, and a "Free-Tail Free-For-All" bat festival.

Tanalgo and Hughes [89] carried out a survey of tourists in Monfort Bat Cave Sanctuary in the Philippines, the world's largest colony of Geoffroy's rousette *Rousettus amplexicaudatus*. They combined a conservation lecture and bat cave-watching and measured tourists' knowledge about bat ecosystem services and willingness to support bat conservation both before and after the events. They found that knowledge about bat ecosystem services significantly increased from 44% to 87% and willingness to support bat conservation from 44% to 61% after the events.

Our findings suggested that support for bat conservation could be promoted among Greek residents by targeting those who disliked bats and sociodemographic groups such as the younger, less educated people, non-farmers, and non-pet owners. Greek residents drew information about bats mostly from obscure, unscientific sources, such as movies, social circles, novels, comics, and the news. These sources usually depict bats in a negative way. The design and implementation of tailor-made environmental education and outreach programs aimed at increasing the factual and experiential knowledge of public groups

about bats and their ecosystem values, based on scientific findings, would improve people's attitudes towards these animals and, subsequently, their intent to contribute to conservation actions [84–89].

## 5. Conclusions

Our findings suggested moderate support and WTP for bat conservation among the Greek population. Emotions, such as likeability (positive) and fear (negative), were the most significant predictors of support for conservation the former and WTP the latter. Older, more educated, farmers and pet owners were the most supportive of bat conservation. The more educated farmers and consumptive recreationists showed the highest WTP. Future research should also examine the effects of other emotions, such as interest, danger, adoration [20,25]. Research should also concentrate on the design, implementation, and assessment of education and outreach programs aimed at improving public attitudes towards bats. The success of such programs would allow for a wider approval of bat conservation and management actions. Our findings would be critical in this process by providing relevant scientific data.

**Author Contributions:** Conceptualization, V.L. and V.J.K.; investigation, O.K.; methodology, V.L., V.J.K., O.K. and A.P.; software, V.L.; validation, V.L. and V.J.K.; formal analysis, V.L. and V.J.K.; resources, V.L., V.J.K., O.K. and A.P.; data curation, V.L. and V.J.K.; writing–original draft preparation., V.L. and V.J.K.; writing–reviewing and editing, V.L., V.J.K., O.K. and A.P.; visualization, V.L.; supervision, V.L. and V.J.K.; project administration, V.L. All authors have read and agreed to the published version of the manuscript.

**Funding:** This research received no external funding.

**Institutional Review Board Statement:** The study was conducted according to the guidelines of the Declaration of Helsinki, and adhered to the ethical standards laid out by the Research and Academic Committee of the International Hellenic University.

**Data Availability Statement:** The data presented in this study are available on reasonable request from the corresponding author.

**Acknowledgments:** We thank survey participants for sharing their time and opinion with us. We also thank four anonymous reviewers whose comments and suggestions helped greatly improve the manuscript.

**Conflicts of Interest:** The authors declare no conflict of interest.

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
