# Peer review of "The Interplay of Likeability and Fear in Willingness to Pay for Bat Conservation"

_2673-4834, doi:10.3390/earth2040046_

Round 1
Reviewer 1 Report
Dear authors first of all congratulations for the effort made trying to better understand an issue that it could be worrying for bat conservationists. I’ve found a very well-crafted introduction that it has been very useful for me. In general, I’ve found that problems with the questionnaire difficult to solve because I think that you created a bias in the result due to the way you asked people. I’ve the feeling that the interviewee knew that the interview was made by a conservationist institution. Besides, I have found few novel information in the manuscript. Some specific comments above:
- Line 114 table od Greece endangered species it is not necessary in the introduction
- To many Hypothesis I feel you have to reduce it to the H1 and explain your results in the discussion
- Line 170. In my opinion this question it is not well proposed because you are predisposing the citizen to answer in favour of bats: “Many of the populations of the 35 bat species of Greece are vulnerable and threatened with extinction, mainly due to the degradation or destruction of mature forests and caves.. Would you support a governmental program to protect bats and their habitats by paying an amount annually for the next five years?”
- Figure 2 it is not necessary
- Table 2. Usually, when you are asking people for delicate issues giving them the possibility of not positioning themselves they move to moderate answers even if don’t feel so.
- Line 263. Almost 60% of the participants dislike bats even they know that the questioner it is made by bat conservationists (according to the question you have made, line 170).
- Line 266. Almost 1/3 of all participants learned about bats from movies. Few or none movies are in favour of bat conservation.
- Line 325. This affirmation (Greece people is willing to pay 4 times more than Us or Mexican) it is too ambitious, in my opinion. When you are making a questionnaire questions are probably the most important methodology factor and you are not comparing questions at all from these different works.
- Lines 369 to 390. In my opinion this part it is too long for the discussion, probably part of it has to be placed in the introduction or removed.
Reviewer 2 Report
An interesting article which addresses a group of species many people consider unappealing, but which are declining bioindicator species, whose loss of habitat reflects wider issues in the natural environment. It is interesting, but not unexpected, to see that people are more willing to pay for the conservation of bats if they consider them appealing/likeable, and not if they find them disgusting or are afraid of them. This is a common issue among the general public and the suggestion of improving outreach and education around these species is very important. The methods and analysis seem sensible and appropriate, but this sort of study is not my personal expertise. I might have used a GLMM instead to incorporate random factors i.e. region, potentially, but if there were none that could be used/collected, the binary logistic approach is good. Through the results, it is unclear what proportions actually answered each question, and how many provided neutral responses (where appropriate), this needs to be included for clarity, and N included if any participants did not respond to specific questions. The discussion needs more emphasis on the implications for science communication, outreach etc. and how that can improve bat conservation.

Reviewer 3 Report
The authors analyse a quite interesting topic with an obvious gap in the literature. The study presents several methodological limitations, however I suggest acceptance at its current form, due to the lack of such studies in Europe and in Greece in particular.
Minor comments
- More discussion is required regarding the sampling protocol. It is unclear from the presentation whether one vilage, town, city was sampled from each of the 13 regions or some other procedure was followed
- The choice of increments in the WTP is incorrect, there shouldn't be round numbers to avoid bias. This unfortunately can't be corrected at this stage, but the choice of the selected value ladder needs further justifications
- It is unclear if the tax is per person or per household
- Level of education variable in Table 2 needs clarification
- The samples' mean reported income is far below Greece's mean income.
- Pet ownerships seems very high, is this representative of the general population?
- It is unclear how the estimated WTP of 21.7 euros was aggregated for the whole population, given that a significant percentage pay zero income taxes, and therefore can't realistically be taxes through an income tax.
Reviewer 4 Report
The paper is an original research work that brings forward a neglected but significant topic; that is, the role of the public in the successful conservation of “unlikable” endangered species such as bats. Moreover, the authors did a great job regarding the theoretical background and the research design in this paper. In addition, they managed to recruit a large and representative cohort (1131 questionnaires were collected) which . Another advantage is that the authors present in great detail the methodology they followed to perform the study and this enables other researchers to replicate the methods. The paper can be published but authors are advised to consider addressing the following comments:
- In line 156, perhaps “surveys” should be replaced with “interviews.
- In line 168, “was” should be deleted.
- The word “following” should be added before the word “option” in line 191.
- Table 5 is placed in the Discussion. However, tables should not be in this part of the paper. I recommend placing this table in the Introduction and comment on it briefly also there in the text.
- Sentence in lines 391-392 is somewhat confusing and should be rephrased.
